



**Drivers of soil C quality and stability: Insights from a topsoil**
**dataset at landscape scale in Ontario, Canada**
Inderjot Chahal[1*], Adam W. Gillespie[2], Daniel D Saurette[2,3], and Laura L. Van Eerd[1]
[1]School of Environmental Sciences, University of Guelph, Ridgetown Campus, Ontario, Canada
[2]School of Environmental Sciences, University of Guelph, Ontario, Canada
[3]Ontario Ministry of Agriculture, Food and Agribusiness, Guelph, Ontario, Canada
Corresponding author: Inderjot Chahal (chahali@uoguelph.ca)
**Abstract**
Although soil C is a critical component of soil health, studies robustly exploring the agronomic
and pedoclimatic effects on soil C are limited, especially at the landscape scale. Therefore, a
dataset of 1511 samples from agricultural fields across Ontario was used to evaluate the impacts
of agronomic and pedoclimatic factors on eight soil C indicators including chemistry and thermal
stability of soil C using the programmed pyrolysis approach. Soil C quality and stability were
largely controlled by the inherent soil characteristics such as soil texture. Significant interactive
effects of cropping system and tillage intensity on soil C indicators were observed; however, the
number of significant effects varied among the three soil textural classes. All soil C indicators
were significantly different among the cropping systems for the coarse textured soils, but the
cropping system differences decreased under medium and fine textured soils. From the pyrolysis
analysis, the hydrogen index (HI) and oxygen index (OI) also confirmed that the soil C chemistry
was influenced by the cropping system. For instance, orchard systems had stable pools of soil C
whereas vegetable systems were associated with less advanced degree of soil C decomposition.
Remaining soil management variables (cover crop use, tillage intensity, and organic amendments)
had less influence on soil C indicators in all soil textural classes. Principal component analysis

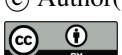



revealed a close association of soil C indicators with the mean annual precipitation (MAP) and
cropping system; suggesting that the quantity and quality of soil C inputs associated with different
cropping systems and increase in precipitation had a large influence on soil C. Our results confirm
the significant effects of agronomic and pedoclimatic variables on chemistry, thermal stability,
and composition of soil C pools, which have long-term implications on soil C storage, mitigating
global climate change, and improving soil health.
**1. Introduction**
Soil C is estimated based on the amount of C in various soil C pools and the transformation
rate of C within these pools due to microbial processes (Cotrufo et al., 2013; Parton et al., 1988).
Several indicators, therefore, have been proposed to assess soil C. Soil organic C (SOC) is one of
the most measured parameters of soil health (Bunemann et al., 2018) and plays a significant role
in many soil functions like nutrient and water cycling and greenhouse gas emissions (Lal, 2016).
Soil organic C comprises C compounds with a wide range of stabilities, and so it takes a long time
(5 to 10 years) to detect changes after implementation of new land use management practices
(Poeplau and Don, 2015). Total organic C includes a labile C pool, which is easily metabolized by
soil microbes and has a rapid turnover time. Measurements of the labile soil C pool, by using the
permanganate oxidizable C (POXC; Weil et al., 2003), potential C mineralization and soil
respiration (Haney et al., 2008) tests, are useful assessments of the quantity of C that is metabolized
by the soil microbes and may offer early detection of SOC changes from land use changes.
Likewise, autoclaved citrate extractable protein (ACE) may be used as a predictor of labile soil N
and C (Agnihotri et al., 2022). Assessment of these soil C indicators, hence, provide direct or
indirect information on the soil C storage and functions across diverse soil textures, management
practices, and climatic conditions (Liptzin et al., 2022). Most of the research on soil C indicators





and its response to management practices is conducted in field experiments and at a small-plot
scale (Chahal et al., 2021; Culman et al., 2013; Mesgar et al., 2024). In this study, we apply these
techniques to samples obtained at the landscape scale from operational agricultural fields. This is
important to comprehensively characterize the relationship between soil C indicators and the farm
management strategies that drive soil health.

Thermal analysis methods, such as programmed pyrolysis, is a novel technique in soil science

and is used to assess the molecular composition and the thermal stability of SOC (Gillespie et al.,
2014). Programmed pyrolysis subjects soil samples to a temperature ramp under an inert
atmosphere and measures the organic and inorganic C released as a function of increasing
temperature (Lafargue et al., 1998; Sebag et al., 2016). The thermal stability is related to the
biodegradation potential of SOC (Peltre et al., 2013; Sebag et al., 2016; Soucémarianadin et al.,
2018), which is inferred from the hydrogen index (HI), oxygen index (OI), and T50 (temperature
at which 50% of the pyrolyzable C has been released). The HI primarily represents fresh,
hydrogenated organic matter, and is related to the labile pool of soil C whereas the OI represents
organic matter that has been oxidized through microbial metabolism, and is a more resistant and
stable pool of soil C (Carrie et al., 2012; Mesgar et al., 2024). The T50 is inversely related to the
decomposition potential, in that increasing T50 indicates lower decomposition potential and thus
more biologically stable organic matter (Gillespie et al., 2014; Gregorich et al., 2015). While the
HI, OI, and T50 are not direct indicators of soil health, the assessment of SOC stability and quality
using these indicators (pyrolysis method) provides valuable knowledge on how to build soil C and
to develop effective strategies to reduce the C loss under different land use management practices.

Soil C quality and stability is controlled by numerous variables such as soil texture, tillage,

crops grown, cover crops, use of organic amendments, changes in temperature, precipitation, and





the interactions among these factors (McDaniel and Grandy, 2016). Furthermore, these controlling
variables influence the composition of SOC and can potentially alter the stable and labile pools of
C (McDaniel and Grandy, 2016; Soon et al., 2007). Adding manure, for instance, to an intensively
managed long-term sorghum-wheat cropping system increased the labile fraction of soil C (Datta
et al., 2018). Soil C mineralization and respiration was increased by using no-tillage, cover crops,
and a diverse crop rotation (Balota et al., 2004; Chahal and Van Eerd, 2020; Viaud et al., 2011).
Adopting reduced tillage along with cover crops or perennial crops increased SOC content and
soil microbial biomass C (Sun et al., 2023). Likewise, POXC increased with the reduction in tillage
intensity and cover cropping (Liptzin et al., 2022). These studies from long-term experiments
confirm that different management practices impact the SOC composition which in turn, affects
the labile and stable pools of soil C. Yet, it is largely unknown if these effects are evident given
the complexity of agricultural fields.

Here, we used a large dataset of mineral topsoil samples collected from agricultural fields

across Ontario through the Ontario Topsoil Sampling Project (OTSP). Previously, the OTSP
dataset was used to assess the soil health scoring functions (Chahal et al., 2023) and SOC:clay
ratio as an indicator of soil functionality (Chahal et al., 2024). The goal of the present study was
to evaluate the impact of agricultural management and environmental variables on soil C indicators
(SOC, 96-h C mineralization potential ($C_{min}$-96h), POXC, Solvita $CO_2$-burst, and ACE) and
indicators of thermal stability of soil C using programmed pyrolysis (HI, OI, and T50). We also
assessed the associations among these soil C indicators at the landscape scale to comprehensively
assess the major drivers of soil C. The study results will contribute to making improved
recommendations regarding selection of soil C indicators and help growers and researchers to
adjust management practices to increase soil C storage.



## 2. Materials and methods

2.1 Soil sample collection

Topsoil samples (n=1511) for this study were collected as a part of the OTSP from 2019 to 2022. The OTSP was a collaborative project between the Ontario Ministry of Agriculture, Food, and Agribusiness (formerly known as Ontario Ministry of Agriculture, Food, and Rural Affairs) and the School of Environmental Sciences at the University of Guelph, to assess the soil physical, chemical, and biological characteristics in agricultural soils in Ontario (Chahal et al., 2023; Chahal et al., 2024). Soil samples were collected from the Ap horizon (median depth of 25 cm), and the sampling depth was terminated at 30 cm. Details about soil sample collection and the selection of the locations is explained in detail in Chahal et al. (2023). Briefly, for each site, three soil samples were collected, georeferenced, and a comprehensive land management survey with the grower was conducted to document information on the crop rotation, type of crops grown, tillage, use of cover crops, and application of organic amendments. Mean annual temperature (MAT) and mean annual precipitation (MAP) data were collected using the WorldClim version 2.1 from 1970 to 2000 (http://www.worldclim.org/data/worldclim21.html). The MAP and MAT were grouped into two (intermediate zone with precipitation between 800 and 1000 mm and wet zone with precipitation greater than 1000 mm) and three classes (0 to 5°C, 5.01 to 10°C, and 10.01 to 15°C), respectively. All the soil samples were classified into three soil textural classes (coarse with sand % ranging between 52 to 94%), medium (between 2 to 78% sand) , and fine (between 1 to 45% sand); Moebius-Clune et al., 2016), five cropping system (annual grain, forage, vegetable, orchard, and perennial), five tillage intensity (conventional tillage, moderate disturbance, light disturbance, no disturbance, and no-tillage), two cover crop (yes or no), and two organic amendment (yes or no) classes. For the tillage intensity classes, conventional tillage represented the moldboard plow

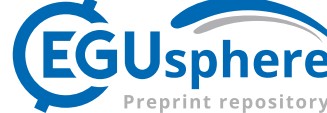

tillage with full soil disturbance, moderate tillage represented more than 2 passes with disk or
chisel plow, light disturbance represented 1 or 2 passes with disk or cultivator, no disturbance
referred to the little or no soil movement as observed in pasture and perennial cropping systems,
and no-tillage represented minimal or no disturbance to the soil such as slot-tillage during planting.
We focused on the mineral soils for this study; thus, 21 soil samples with topsoil SOC
concentration more than 8.7% were removed from the dataset (Chahal et al., 2024). Therefore,
total number of samples used for the soil C analysis were 1490. It is important to note that the
number of observations for each soil C indicator (given in section 2.2) varied by the indicator and
hence the management practices. For instance, programmed pyrolysis was restricted to 151 soil
samples selected via conditioned Latin hypercube approach (Minasny and McBratney, 2006) as
representative of full dataset (Chahal et al., 2023).
2.2. Laboratory analyses
After QA/QC analysis, the database consisted of eight soil C indicators: SOC (n=1490), $C_{min}$-
96h (n=1017), POXC (n=1413), Solvita $CO_2$-burst (n=768), ACE protein (n=151), HI (n=151), OI
(n=151) and T50 (n=151). Soil organic C concentration was calculated as the difference between
total C and inorganic C. Total C was estimated using the dry combustion method (samples were
combusted at 1300 °C) on LECO 828 Series CN analyzer (Skjemstad et al., 2008). Inorganic C
was determined by subjecting a ground subsample of soil to combustion in a muffle furnace at
470°C to remove organic C (Krom and Berner, 1983). Soil C mineralization ($C_{min}$-96h) was
quantified using the KOH trap method and was determined by measuring the concentration of $CO_2$
evolved (mg $CO_2$-C 20 $g^{-1}$ soil) when 7.5 mL water was added to 20 g air dried soil placed in an
air-tight jar with 9 mL of 0.5 M KOH at room temperature (Schindelbeck et al., 2016).
Permanganate oxidizable C ($\mu g\ g^{-1}$) was measured using spectrophotometer where the colorimetric





change due to the reduction of manganese in a potassium permanganate solution on air dried soil
was quantified (Moebius-Clune et al., 2016). Solvita $CO_2$-burst was quantified as per the updated
protocol by the Woods End® Laboratories Inc., Mt. Vernon, ME. A 30-cc scoop (around 25 to 35
g) of oven-dried (40°C) soil was placed in a vial and was wetted using 9 to 10 mL distilled water.
The vials were transferred to 475 mL glass jars and Solvita $CO_2$-burst paddles were inserted into
each jar. The jars were sealed and left undisturbed for 24 h at room temperature and evolved $CO_2$
concentration (mg $kg^{-1}$) was determined using a digital colorimeter reader (Brinton, 2019).
Autoclaved citrate extractable protein (mg $kg^{-1}$) was quantified on air-dried soils by autoclaving,
centrifuging, and treating a sodium citrate soil extract with bicinchoninic acid (Schindelbeck et al.,
2016). Soil particle size analysis was done using the pipette method where soil organic matter was
removed by treating the soil with hydrogen peroxide (Sheldrick and Wang, 1993). Consistent with
soil textural grouping recommended by Moebius-Clune et al. (2016), the dataset was divided into
three soil textural classes (coarse, medium, and fine).

The programmed pyrolysis analysis was conducted using a HAWK pyrolyzer (Wildcat

Technologies, Humble, Texas, USA) at the Canadian Geological Survey in Calgary Alberta
according to Gillespie et al. (2014) and Gregorich et al. (2015). About 70 mg air-dried ground soil
sample is subjected to a constant temperature of 300°C for 3 min under helium gas, to quantify
free hydrocarbons using a flame ionization detector (mg HC $g^{-1}$) and referred to as S1. Next, the
soil sample was heated at a rate of 25°C per minute until 650°C where hydrocarbons were released
(i.e., cracking SOC) and quantified as S2. The concentration of $CO_2$ (mg CO2 $g^{-1}$) released during
S1 and S2 was determined using an infrared detector and represented as S3. Hydrogen index of a
soil sample represents the ratio of all hydrocarbons measured as S1+S2 divided by SOC, whereas
OI refers to the ratio of $CO_2$ released (S3) divided by SOC (Lafargue et al., 1998). The stability of



soil C is represented by the T50 which is the temperature at which 50% of the SOC was pyrolyzed
(Gillespie et al., 2014; Gregorich et al., 2015).
2.3 Data analysis
All soil data were analyzed using the SAS (SAS Institute version 9.4, Cary, NC, USA). A
variance component analysis was conducted to test the relative importance of the agronomic and
pedoclimatic variables for each soil C indicator. For each soil textural class, the main as well the
interactive effect of the agronomic management practices (cropping system, tillage intensity, cover
crops, and organic amendments) on soil C indicators were tested using PROC GLIMMIX in SAS.
The relationship among the soil C indicators was tested using the Pearson correlation analysis
(PROC CORR). To linearize the relationships among indicators, all the indicators were log-
transformed for correlation analysis. Consistent with Liptzin et al. (2022) and Mesgar et al. (2024),
a principal component analysis using PRINCOMP procedure was also conducted to explore how
the soil C indicators interacted with the site characteristics (i.e., agronomic management practices
(cropping system, tillage, cover crop, organic amendment) and pedoclimatic conditions (sand, silt,
clay, mean annual temperature and precipitation)). The statistical significance of all the tests was
assessed at $P<0.05$.
**3. Results and discussion**
3.1 Agronomic and pedoclimatic effects on soil C indicators
The results of variance partitioning revealed that soil textural classes and cropping system
explained a larger percentage of variance for all soil C indicators compared to the other variables
(tillage intensity, cover crop, organic amendments, MAP and MAT; Table 1). For SOC, the amount
of variance explained by soil texture was higher than for cropping system (60.5% and 26.4%,
respectively), and for POXC, it was 53.5% and 21.7%, respectively (Table 1). Interestingly, the



percentage of variance explained by texture was comparable or less than for cropping system in
$C_{min}$-96h (38.8% and 36.9%), Solvita $CO_2$-burst (37.5% and 38.7%), and ACE (4.87% and 50.7%)
(Table 1). The type of main crops grown (i.e., the cropping system) was found to be a key driver
of labile pools of soil C (such as $C_{min}$ and ACE) in a study by Amsili et al. (2021). Soil texture
(75.2%) also explained a large amount of variance in the thermal-based parameters of soil C (Table
1). Measures of SOC stability (T50) and quality (HI and OI) were largely controlled by soil texture
for T50 and cropping system for HI and OI. Unlike cropping system and soil textural classes,
tillage intensity was not found to be an important predictor of any of the tested soil C indicators
(Table 1). The least amount of variance in soil C indicators was explained by MAT, use of cover
crops, and organic amendments (Table 1). Therefore, consideration of soil textural class and
cropping system is needed when interpreting the soil C indicators (Nunes et al., 2021).
Table 2 shows the variance analysis on soil C indicators broken out by texture class and
parameter, and Table 3 shows the mean values and groupings. In all soil textural classes, significant
differences in SOC, $C_{min}$-96h, and POXC concentration due to cropping systems were observed
(Table 2). The greatest SOC concentrations were observed under perennials and when forages were
grown with annual crops in all soil textural classes (Table 3). Likewise, forages and perennial
systems had greater or comparable concentrations of $C_{min}$-96h and POXC than the remaining
systems in all soil textural classes (Table 3). Less soil disturbance due to tillage, greater diversity
of crop species, continuous presence of living roots in perennial systems perhaps contributed to
the greatest concentration of SOC, $C_{min}$-96h, and POXC observed (Amsili et al., 2021; Congreves
et al., 2015; Nunes et al., 2020). Compared to annual grain, perennial and forage systems provide
a more temporally consistent (i.e., no fallow period) source of substrate quality to microbial
communities mainly due to high C:N ratio, lower lignin content, and high concentration of



mineralizable C (Mesgar et al., 2024). For most of the soil C indicators, annual grain, orchard, and
vegetable cropping systems had the lowest soil C (Table 3). Vegetable and annual grain systems
are intensively managed with high intensity of tillage and have lower soil C inputs (Norris and
Congreves, 2018; Nunes et al., 2020), which negatively impact soil C. Therefore, in all soil textural
classes, diversification of cropping systems and addition of organic amendments are critical
components of building soil C. Similarly, studies by Adhikari and Hartemink (2017) and Presley
et al. (2004) demonstrated that adoption of conservational agricultural practices such as reduced
tillage, diversified cropping systems and addition of organic amendments contributed to a build up
of SOC even on coarse textured soils.
While T50 was not statistically different due to cropping system and tillage practices in all soil
textures (Table 2), annual grain had the highest whereas forage had the lowest T50 across all soil
textures (Table 3 and Figure 1a). These results suggest that annual grain perhaps contribute to more
resistant organic matter additions to soil whereas forage systems might add relatively easily
decomposable residue. Usually, forage cropping systems (specifically legumes) have a higher
residue quality (high biomass N and low C:N) and result in larger proportion of labile fractions of
soil C. Mesgar et al. (2024) found similar results where crop rotations with forages (such as alfalfa
or red clover) contributed to labile components of soil C. Furthermore, fine textured soils had the
highest whereas coarse and medium textured soils had the lowest T50 (Table 3 and Figure 1b),
confirming that the soil texture and clay content influence the thermal stability of soil organic
matter.
The other thermal-based parameters of soil C characterization were the HI and OI, which
represented the maturity level of soil organic matter. Typically, a high HI represents a thermally
labile pool of organic matter which is enriched with hydrogen and the freshly added carbohydrates





and lignin (Mesgar et al., 2024). Conversely, OI represents a more resistant pool of organic matter
following the oxidation processes occurring during the soil organic matter decomposition (Carrie
et al., 2012; Mesgar et al., 2024; Saenger et al., 2013). Simultaneous reduction in both the HI and
OI indicates aromatization. The HI was not different among the agronomic management practices
in coarse and fine textured soils, but significant differences were observed in medium textured
soil. Among the cropping systems in medium textured soils, perennial (200 mg HC g$^{-1}$ OC) had
greatest while annual grain (132 mg HC g$^{-1}$ OC) had the least HI (Table 3). These results confirm
that the diversified cropping systems with forages and perennial crops had a higher quantity and
quality of H-rich labile components of soil organic matter and represented a more labile state of
organic matter decomposition than the intensively managed systems with less C inputs (Ding et
al., 2006; Mesgar et al., 2024). Furthermore, differences in OI due to cropping system were
detected in coarse textured soil only (Table 2). Orchard (203 mg $CO_2$ g$^{-1}$ OC) had greatest whereas
vegetable (147 mg $CO_2$ g$^{-1}$ OC) had the least OI in coarse textured soils; suggesting that the orchard
systems represent a more advanced state of soil organic matter decomposition than the other
cropping system categories.

A significant interaction between cropping system and tillage intensity was detected for some

of soil C indicators in all soil textural classes (Table 2), but trends varied among indicators and
texture. For the thermal based soil C indicators, significant interaction between cropping system
and tillage intensity was observed for the HI in medium-textured soil and for OI in coarse textured
soil (Figure 2, Table S1 to S4). Given that the annual grain and forage cropping systems showed
the strongest contrast for these indicators with sufficient number of observations, we selected only
these two cropping systems to evaluate the effects of tillage intensity (Figure 2, Table S1). In
coarse-textured soils, $C_{min}$-96h, POXC and Solvita $CO_2$-burst concentrations were significantly



impacted by the intensity of tillage adopted in annual grain and forage systems, while the
remaining indicators were comparable across all the tillage and cropping system combinations
(Table S1). In medium-textured soils, all soil C indicators except Solvita $CO_2$-burst had a
significant interaction between cropping system and tillage intensity (Table S1). In fine textured
soils, all but ACE were significantly different among the cropping system and tillage treatment
combinations (Table S1). Therefore, medium and fine textured soils had a greater number of
interactions than coarse textured soils. Overall, the significant interaction of cropping system with
tillage demonstrates that the changes in soil C pools brought on by various tillage treatments is
dependent on the type of the crop species grown and the soil texture. Similar interactive effects of
tillage and cropping system on soil health were reported by Angon et al. (2023).
A pseudo-Van Krevelen diagram was created by plotting HI against OI (Figure 3) to visually
characterize the composition of soil organic matter across various cropping systems (Carrie et al.,
2012; Mesgar et al., 2024). We found that most of the orchard systems have oxygenated products
and represented the more stable pool of soil organic matter whereas for vegetable systems, the
organic matter composition mainly consisted of hydrogenated products and was associated with a
less advanced stage of organic matter decomposition (Figure 3). For the remaining systems, points
found closer to the origin, such as in the annual grain site data, suggests that organic matter
structures are undergoing aromatization processes compared to sites with forages or with perennial
crops.
It is important to note that the frequency count of observations within the various tillage
intensity classes among the cropping system categories was not equal nor balanced (Figure 4) but
are reflective of typical management practices employed within the various cropping systems. For
instance, and as expected, the number of observations collected from the 'no disturbance' category





was greatest in perennial systems (n=142, Figure 4). Vegetable (n=5) and orchard (n=4) cropping
systems had the least number of observations for no-tillage (Figure 4). Orchards had the least
number of total observations (n=33, Figure 4), which is consistent with Ontario agriculture census
data where orchards represent 7.03% of farmland (Fruit and Vegetable Survey, Statistics Canada
2023). Furthermore, the frequency count of samples collected from coarse textured soils (n=308)
was lower than medium (n=642) and fine textured soils (n=540, Figure 5) which is largely
attributed to glacier deposits that shaped the region and the topography where more sand on top
and less fine particles as they are more prone to loss due to erosion. While clearly reflective of
Ontario soils and agriculture, the discrepancy in the count of observations suggests the need to be
cautious in directly attributing the results to a system.
3.2 Relationships of soil C indicators with agronomic and pedoclimatic variables
Principal component analysis was conducted where first and second PCs explained 32% and
17% of the variance in the data, respectively, which is consistent with variance explained in other
studies (Liptzin et al., 2022). Based on the PCA, soil C indicators and the measures of SOC quality
(HI and OI) were closely associated with the cropping system, MAP, and organic amendments,
and were negatively associated with MAT (Figure 6), suggesting a higher value for soil C
indicators under cooler temperatures. Although precipitation and organic amendments did not
explain a large amount of variance in our dataset (Table 1), the PCA demonstrated that the soil C
indicators increased with an increase in precipitation (Figure 6). Climate has been a key
determinant of soil C (Jenny, 1941). The interactions among the soil microbes, crop residues, and
plant root exudates mainly control the influence of climatic variables on soil C (Schmidt et al.,
2011). Our results of high values of soil C indicators with an increase in MAP and decrease in
MAT were consistent with studies conducted in North America (Burke et al., 1989; Liptzin et al.,



2022) and globally (Jobbagy and Jackson, 2000). While relationships of soil C indicators with
temperature and precipitation were consistent with expectations, it was surprising to see an effect
given the relative minimal differences within the province. In Ontario agricultural zones, MAT
varies by only $1^0$C, and MAP by approx. 100 mm. Given current climate change predictions, these
results have powerful implications for C sequestration and soil functioning under future climate
conditions.

Furthermore, silt and clay content were grouped together in PCA and were positively associated

with the soil C indicators in the second axis, whereas sand content was negatively associated with
the indicators in the second axis of the PCA (Figure 6). Typically, soils rich in clay content have
higher C retention capacity than coarse textured soil (von Lutzow et al., 2006); hence, a positive
association was observed between clay content and soil C indicators in our study (Congreves et
al., 2015). Although important, the relationship between soil texture and soil C indicators
(particularly POXC and respiration) has not been explored enough in the literature (Nunes et al.,
2020; Sinsabaugh et al., 2008).

Soil C indicators and HI were negatively associated with tillage intensity and cover crops on

the first PC axis (Figure 6). Increase in tillage intensity reduces soil C (mainly the topsoil C) by
increasing the mineralization of soil organic matter, disrupting the soil structure, and decreasing
soil microbial populations and communities (Nunes et al., 2020). Therefore, adopting reduced or
minimum tillage practices might contribute to building soil C and help to mitigate the negative
impacts of climate change. Numerically greater SOC concentration observed with reducing tillage
intensity in our study suggests the potential of the sustainable land use management practices on
sequestering C, reducing $CO_2$ emissions, and mitigating the global warming effect (Melland et al.,
2017) and greenhouse gas emissions (Mangalassery et al., 2014).



Except POXC in fine-textured soils, we did not find a significant effect of tillage intensity on

soil C indicators (Table 2 and S5). It is important to note that in our study, the participants were
asked to choose one of the tillage intensity categories, which might have caused a variability in
the recorded data (i.e. descriptive terms and actual disturbance on the farm) and might not be a
true reflection of tillage intensity over the long-term. Moreover, the producer interpretation of the
tillage intensity categories could have added uncertainty. Additionally, negative association of soil
C indicators with cover crops and a very small response of indicators to cover crops (Figure 6 and
Table S6) is not clear but might be attributed to a smaller number of observations with cover crops
in our study (n=55). The cover crop effects on soil C indicators are largely dependent on the cover
crop management factors such as type of cover crop species grown, frequency and duration of
cover cropping, planting date, and termination time of cover crops (Blanco-Canqui et al., 2015;
Peng et al., 2024). Due to the unavailability of the cover crop management factors in our study, the
interpretation of cover crop effect is challenging but clearly demonstrates that other management
factors (cropping system and tillage system) have a greater effect on soil C indicators.

3.3. Relationships among the soil C indicators

To better understand associations among soil C indicators, correlation analysis was conducted

on the soil C indicators and the indicators characterizing the chemical composition of soil organic
matter (Table 4). Interestingly, despite having a range of soil textural classes and cropping system
categories, strong positive significant relationships were observed among the soil C indicators
(Table 4). Among the indicators, SOC and POXC had the strongest positive relationship (r=0.81),
which was consistent with Liptzin et al. (2022) and Culman et al. (2012). Consistent with Amsili
et al. (2021) and Nunes et al. (2020), our results confirm that an increase in SOC positively impacts
the soil microbial activity (as demonstrated by Solvita $CO_2$-burst and $C_{min}$-96h) and the quality of





soil organic matter (as demonstrated by POXC and ACE). Significant moderate negative
associations (r=-0.24 to -0.35) were observed between HI and OI, HI and T50 (Table 4). Consistent
with Mesgar et al. (2024), our results suggest that the indicators representing the labile components
of soil organic matter such as HI were negatively associated with OI (an indicator of resistant pool
of soil organic matter) and T50. We also observed that the indicators defining the stable and labile
pools of soil C via the thermal analysis had a positive relationship with the soil C indicators such
as SOC and C mineralization (Table 4). The correlation analysis also confirmed that the easily
decomposable component of soil C (e.g., HI) was closely related to the labile indicators of soil C
such as soil respiration and ACE (Table 4). Collectively, these results confirm the efficacy and
applicability of the programmed pyrolysis method as a valuable tool to study the biochemical
composition and decomposition potential of soil C.
**4. Conclusions**
This study focused on understanding the key drivers of soil C quality and stability in agricultural
production systems at a landscape scale. To the best of our knowledge, this is the first study to
evaluate the agronomic and pedoclimatic effects on the measures of soil C quality and C stability
(particularly with the pyrolysis approach) in North America at the landscape scale. Our results
revealed that soil textural classes and cropping system had a strong influence on both quality and
stability of soil C indicators evaluated in this study. The cropping system differences on soil C are
mainly related to the quantity and quality of residue C inputs, which in turn are primarily dependent
on the cropping system, cover crops, tillage intensity, and organic amendments. Among the
cropping systems, we found a greater concentration of soil C in forage and perennial cropping
systems than the annual grain and vegetable systems. Likewise, forage cropping systems had a
greater preservation of labile components while orchards had a more stable pool of soil C. Our



results, therefore, confirmed that agricultural management-induced factors play a crucial role in
understanding the chemical composition of soil organic matter. The indicators representing the
labile pools of soil C (such as POXC and $C_{min}$-96h) were positively correlated with the parameters
indicating readily decomposable fractions of soil C (i.e., HI).
Furthermore, all indicators had a positive association with precipitation and a negative
relationship with temperature suggesting an increase in the indicator values at cool and wet
conditions despite low differences in values. Increase in the tillage intensity also negatively
impacted the soil C indicators. Overall, the management-induced differences in soil C indicators
in our study imply the benefits of adopting sustainable agricultural practices on building soil C,
reducing $CO_2$ emissions, and mitigating the negative effects of global climate change. While the
findings of our study suggest a significant impact of agronomic and pedoclimatic variables on the
soil C, it is important to note that the results of this study pertain only to the topsoil. It is possible
that the same trends or effects on soil C quality and stability might not be observed at the deeper
soil depths; hence, suggesting a need for future research.
**Code and data availability**
Data will be made available upon request but is not available in an online repository to protect
the privacy of the participants in this project.
**Supplement**
The supplementary tables and figures related to this article are attached along with the
manuscript text.
**Author contributions**
IC, writing, data analysis, data interpretation and presentation, and editing the manuscript; DD,
data organization and curation, reviewing and editing the manuscript; AW, funding acquisition,



supervising, reviewing, and editing the manuscript; LVE, funding acquisition, supervising,
reviewing, and editing the manuscript.
**Competing interests**
The authors declare that they have no conflict of interest.
**Acknowledgments**
Authors are grateful to the financial support provided by Ontario Ministry of Agriculture, Food
and Agribusiness (OMAFA) and the Ontario Agri-Food Innovation Alliance, a collaboration
between the Government of Ontario and the University of Guelph. We would like to thank the
undergraduate student Jane Bellefleur who conducted the lab analysis for ACE.

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





**Table 1** Partitioning of variance of soil C indicators[z] in the Ontario Topsoil Sampling Project database from 2019 to 2022 into soil and crop management, soil texture, and climatic variables.

| | | SOC | $C_{min}$-96h | POXC | Solvita $CO_2$-burst | ACE | T50 | HI | OI |
|---|---|---|---|---|---|---|---|---|---|
| Variables | df | | | | | | | | |
| Cropping system | 4 | **26.4** | **36.9** | **21.8** | **38.8** | **50.7** | 10.0 | **59.8** | **43.0** |
| Tillage intensity | 4 | 4.82 | 18.0 | 6.30 | 11.8 | **22.3** | **10.4** | 11.0 | 10.7 |
| Use of cover crop | 1 | 0.23 | 2.61 | 1.07 | 5.07 | 13.3 | 4.21 | 1.02 | 2.14 |
| Use of organic amendment | 1 | 0.55 | 0.67 | 10.8 | 6.22 | 5.68 | 0.01 | 3.94 | 5.54 |
| Soil textural classes | 2 | **60.5** | **38.8** | **53.5** | **37.5** | 4.87 | **75.2** | **24.1** | **35.6** |
| Mean annual precipitation | 1 | 7.35 | 2.63 | 6.52 | 0.11 | 0.98 | 0.43 | 0.02 | 1.70 |
| Mean annual temperature | 1 | 0.15 | 0.39 | 0.01 | 0.59 | 1.82 | 0.84 | 0.01 | 1.22 |
| Model $R^2$ | | 0.16 | 0.28 | 0.13 | 0.22 | 0.23 | 0.37 | 0.19 | 0.21 |

[z] Number of observations used were: SOC= 1490, $C_{min}$-96h=1017, Solvita $CO_2$-burst=768, and POXC=1413, whereas the number of observations for ACE and programmed pyrolysis parameters (HI, OI, and T50) were 151.

SOC=soil organic carbon; $C_{min}$-96h= 96-hr carbon mineralization; POXC=permanganate oxidizable carbon; Solvita $CO_2$-burst= 24 h soil respiration test; ACE=autoclaved citrate extractable protein index; HI=hydrogen index; and OI=oxygen index.

For each soil C indicator, bold font indicates the top two variables explaining the variance.



**Table 2** Within each soil textural class, variance analysis (*P* values) of soil and crop management
parameters on soil C indicators[z] analyzed in the Ontario Topsoil Sampling Project from 2019 to 2022.

| | | SOC | $C_{min}$-96h | POXC | Solvita $CO_2$-burst | ACE | T50 | HI | OI |
|---|---|---|---|---|---|---|---|---|---|
| | | mg g⁻¹ | mg $CO_2$-C 20 g⁻¹ soil | µg g⁻¹ | mg kg⁻¹ | mg kg⁻¹ | °C | mg HC g⁻¹ OC | mg $CO_2$ g⁻¹ OC |
| | | | | | | *P* values | | | |
| Parameter | df | | | | | Coarse- textured | | | |
| Cropping system[y] | 4 (3) | **0.0403** | **0.0003** | **0.0044** | **0.0077** | **0.0298** | 0.8275 | 0.5131 | **0.0109** |
| Tillage intensity | 4 | 0.4035 | 0.6495 | 0.9057 | 0.4571 | 0.0505 | 0.7480 | 0.7405 | 0.6670 |
| Cover crop | 1 | 0.5625 | 0.2332 | 0.4637 | 0.3627 | **0.0055** | **0.0129** | 0.0887 | 0.4202 |
| Use of organic amendment | 1 | 0.9626 | 0.6656 | 0.1358 | 0.9150 | 0.6184 | 0.1869 | 0.3257 | 0.4489 |
| Crop x Tillage | 16 (12) | 0.1535 | **0.0059** | **0.0071** | **0.0144** | 0.4207 | 0.4651 | 0.9249 | **0.0066** |
| | | | | | | Medium-textured | | | |
| Cropping system | 4 | **0.0020** | **<0.0001** | **0.0003** | 0.7583 | 0.3830 | 0.4477 | **0.0005** | 0.3245 |
| Tillage intensity | 4 | 0.7602 | 0.2161 | 0.1347 | 0.0674 | 0.6528 | 0.5661 | 0.0595 | 0.2743 |
| Cover crop | 1 | 0.9172 | **0.0091** | 0.8947 | 0.2020 | 0.8529 | 0.6813 | 0.1646 | **0.0123** |
| Use of organic amendment | 1 | 0.7148 | 0.0735 | 0.4686 | 0.2045 | 0.6543 | 0.5044 | 0.0555 | 0.9541 |
| Crop x Tillage | 16 | **0.0025** | **<0.0001** | **0.0025** | 0.0610 | **0.0298** | 0.5861 | **0.0105** | 0.5871 |
| | | | | | | Fine-textured | | | |
| Cropping system[y] | 4 (3) | **0.0018** | **<0.0001** | **0.0485** | **<0.0001** | 0.1199 | 0.8067 | 0.2488 | 0.9702 |
| Tillage intensity | 4 | 0.7322 | 0.7080 | **0.0210** | 0.1948 | 0.4784 | 0.9968 | 0.5926 | 0.5507 |
| Cover crop | 1 | 0.1991 | 0.4240 | 0.0962 | 0.1037 | 0.1585 | 0.3945 | 0.8834 | 0.9502 |
| Use of organic amendment | 1 | 0.8311 | 0.2033 | 0.1543 | 0.5495 | 0.0905 | 0.0739 | 0.3373 | 0.3781 |
| Crop x Tillage | 16 (12) | **0.0006** | **<0.0001** | **0.0007** | **<0.0001** | 0.2743 | 0.5131 | 0.6867 | 0.3253 |

Bold font indicates statistically significant treatment differences at *P*<0.05.
[z] Number of observations used were: SOC= 1490, $C_{min}$-96h=1017, Solvita $CO_2$-burst=768, and POXC=1413, whereas the number of observations for ACE and
programmed pyrolysis parameters (HI, OI, and T50) were 151.
[y] For the coarse and fine-textured soil, number of cropping system categories analyzed for HI, OI, and T50 were 4. For instance, no soil samples were collected from
the perennial cropping system in coarse-textured soil and from vegetable system in fine-textured soils. Hence, the degree of freedom for cropping system and
interaction between cropping system and tillage treatments was adjusted accordingly in the statistical model.
SOC=soil organic C; $C_{min}$-96h= 96-hr carbon mineralization; POXC=permanganate oxidizable carbon; Solvita $CO_2$-burst= 24 h soil respiration test; ACE=autoclaved
citrate extractable protein index; HI=hydrogen index; and OI=oxygen index.




**Table 3** Within each soil textural class, mean (SE) values of the soil C indicators[z] by the cropping system category sampled in the Ontario Topsoil Sampling Project from 2019 to 2022.

| | | SOC | $C_{min}$-96h | POXC | Solvita $CO_2$-burst | ACE | T50 | HI | OI |
|---|---|---|---|---|---|---|---|---|---|
| Cropping system | n | mg g$^{-1}$ | mg $CO_2$-C 20 g$^{-1}$ soil | µg g$^{-1}$ | mg kg$^{-1}$ | mg kg$^{-1}$ | °C | mg HC g$^{-1}$ OC | mg $CO_2$ g$^{-1}$ OC |
| | | | | | Coarse-textured | | | | |
| Annual grain | 183 | 16.9ab (0.70) | 14.2bc (0.96) | 473ab (16.0) | 53.1b (3.35) | 5.49ab (0.78) | 416 (1.58) | 152 (5.94) | 168bc (4.26) |
| Forage | 53 | 19.4a (1.20) | 20.4a (1.49) | 547a (25.4) | 72.0a (6.39) | 8.45a (1.47) | 412 (4.19) | 163 (15.7) | 207a (11.2) |
| Vegetable | 32 | 15.4ab (1.50) | 10.9c (1.61) | 441ab (31.6) | 89.4a (12.2) | 6.60ab (1.75) | 415 (5.13) | 162 (19.2) | 147c (13.8) |
| Orchard | 9 | 11.7b (2.70) | 14.4abc (5.70) | 342b (46.0) | 51.4ab (9.23) | 4.71b (1.75) | 419 (5.13) | 132 (19.2) | 203ab (13.8) |
| Perennial | 31 | 19.4ab (1.70) | 18.5ab (1.78) | 510ab (34.5) | 53.6ab (16.1) | -- | -- | -- | -- |
| All | 308 (29)[z] | 17.2 | 15.6 | 478 | 61.4 | 6.80 | 415 | 154 | 172 |
| | | | | | Medium-textured | | | | |
| Annual grain | 335 | 20.8b (0.40) | 20.1b (0.57) | 591a (9.74) | 68.3 (2.76) | 5.55ab (0.43) | 421 (1.65) | 132b (5.98) | 196 (5.20) |
| Forage | 129 | 20.2b (0.60) | 22.6ab (0.92) | 584a (15.2) | 74.4 (3.66) | 6.72ab (0.50) | 417 (2.24) | 159ab (8.12) | 195 (7.07) |
| Vegetable | 54 | 20.0b (1.00) | 19.7b (1.27) | 575a (22.3) | 65.4 (7.74) | 4.46ab (1.02) | 415 (5.72) | 134ab (20.7) | 164 (18.0) |
| Orchard | 20 | 16.6b (1.50) | 17.7b (2.58) | 439b (34.4) | 81.4 (7.94) | 7.63ab (1.32) | 420 (5.72) | 184ab (20.7) | 180 (18.0) |
| Perennial | 104 | 24.0a (0.80) | 25.2a (1.03) | 560a (18.7) | 81.1 (6.88) | 10.1a (1.38) | 412 (3.61) | 200a (13.1) | 188 (11.4) |
| All | 642 (46) | 21.1 | 21.6 | 581 | 72 | 6.30 | 419 | 149 | 193 |
| | | | | | Fine-textured | | | | |
| Annual grain | 315 | 23.1b (0.50) | 21.2c (0.68) | 574c (10.7) | 76.2c (2.20) | 5.62 (0.34) | 428 (1.11) | 132 (4.23) | 186 (3.14) |
| Forage | 131 | 25.5ab (0.80) | 25.3ab (1.05) | 623ab (15.8) | 86.9b (3.36) | 5.93 (0.64) | 427 (2.46) | 140 (9.38) | 191 (6.97) |
| Vegetable | 15 | 22.5b (2.50) | 21.6bc (2.38) | 606b (47.9) | -- | -- | -- | -- | -- |
| Orchard | 4 | 25.6ab (4.20) | 35.0a (3.99) | 694a (80.8) | 107ab (13.6) | 9.02 (2.05) | 426 (8.16) | 161 (31.3) | 195 (23.1) |
| Perennial | 72 | 28.9a (1.30) | 30.5a (1.56) | 613ab (26.7) | 108a (5.81) | 7.91 (0.82) | 427 (2.58) | 161 (9.84) | 202 (7.31) |
| All[y] | 540 (76) | 24.5 | 23.4 | 603 | 83.9 | 6.27 | 427 | 138 | 189 |

[z] The number in the parenthesis represents the number of observations for ACE and the programmed parameters (T50, HI, and OI) in each soil textural class.

[y] There were 3 observations within the fine-textured soils for which cropping system details were missing.

SOC=soil organic carbon; $C_{min}$-96h= 96-hr carbon mineralization; POXC=permanganate oxidizable carbon; Solvita $CO_2$-burst= 24 h soil respiration test; ACE=autoclaved citrate extractable protein index; HI=hydrogen index; and OI=oxygen index.; -=not applicable.





**Table 4** Pearson correlation coefficients (r) among soil C indicators sampled in the Ontario Topsoil Sampling
Project from 2019 to 2022.

| Indicators^ | SOC | $C_{min}$-96h | POXC | Solvita $CO_2$-burst | ACE | HI | OI | T50 |
|---|---|---|---|---|---|---|---|---|
| | mg g$^{-1}$ | mg $CO_2$-C 20 g$^{-1}$ soil | µg g$^{-1}$ | mg $CO_2$-C kg$^{-1}$ | mg kg$^{-1}$ | mg HC g$^{-1}$ OC | mg $CO_2$ g$^{-1}$ OC | °C |
| SOC | 1.00 | | | | | | | |
| $C_{min}$-24h | 0.67*** | 1.00 | | | | | | |
| POXC | 0.81*** | 0.63*** | 1.00 | | | | | |
| Solvita $CO_2$-burst | 0.36*** | 0.46*** | 0.34*** | 1.00 | | | | |
| ACE | 0.72*** | 0.58*** | 0.66*** | 0.17** | 1.00 | | | |
| HI | 0.28** | 0.42*** | 0.24** | NS | 0.59*** | 1.00 | | |
| OI | 0.21** | 0.27** | NS | NS | NS | -0.30** | 1.00 | |
| T50 | NS | NS | NS | NS | NS | -0.24** | NS | 1.00 |

SOC=soil organic carbon; $C_{min}$-96h= 96-hr carbon mineralization; POXC=permanganate oxidizable carbon; Solvita $CO_2$-burst= 24 h soil respiration test;
ACE=autoclaved citrate extractable protein index; HI=Hydrogen Index; OI=Oxygen Index; and NS=Non-significant.
n=1490 for soil organic C; n=1017 for $C_{min}$-96; n=1413 for POXC; n=768 for Solvita $CO_2$-burst; n=151 for ACE, HI, OI, T50.
**,*** indicates statistical significance at $P<0.05$ and $P<0.0001$, respectively.
^All indicators were log-transformed prior to analysis.



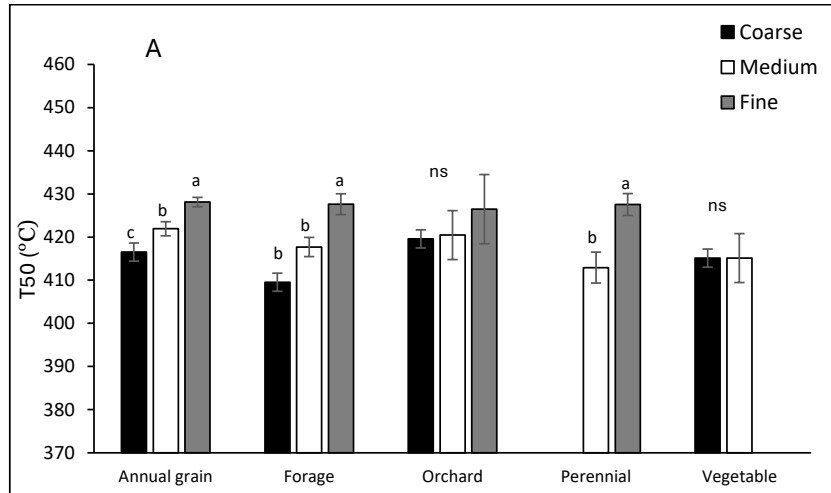


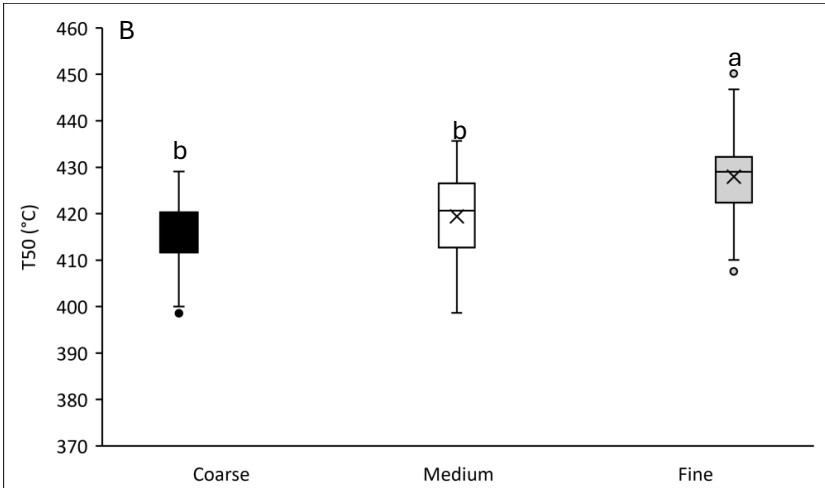


**Figure 1** Plots of T50 demonstrating differences due to soil textural class within each cropping system category (A)
and soil textural classes (B) for the soils in the Ontario Topsoil Sampling Project in 2019 (n=151). Different letters
indicate statistically significant differences at *P*<0.05. ns represents non-significant statistical differences among soil
textural classes within the cropping system category.






**Figure 2** Box plots demonstrating the interactive effects of cropping system category and tillage intensity on soil C indicators sampled in the Ontario Topsoil Sampling Project from 2019 to 2022. SOC=soil organic carbon; $C_{min}$-96h= 96-hr carbon mineralization; POXC=permanganate oxidizable carbon; Solvita $CO_2$-burst= 24 h soil respiration test; ACE=autoclaved citrate extractable protein index. Due to insufficient number of observations, data for other cropping systems not shown.




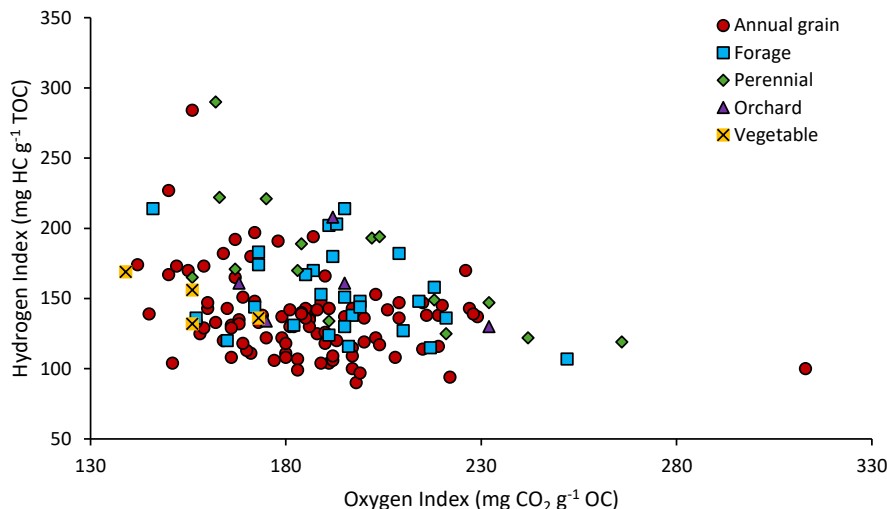

**Figure 3** Pseudo van Krevelan diagram from programmed pyrolysis data for soils indicating cropping system category sampled in the Ontario Topsoil Sampling Project in 2019 (n=151).



585

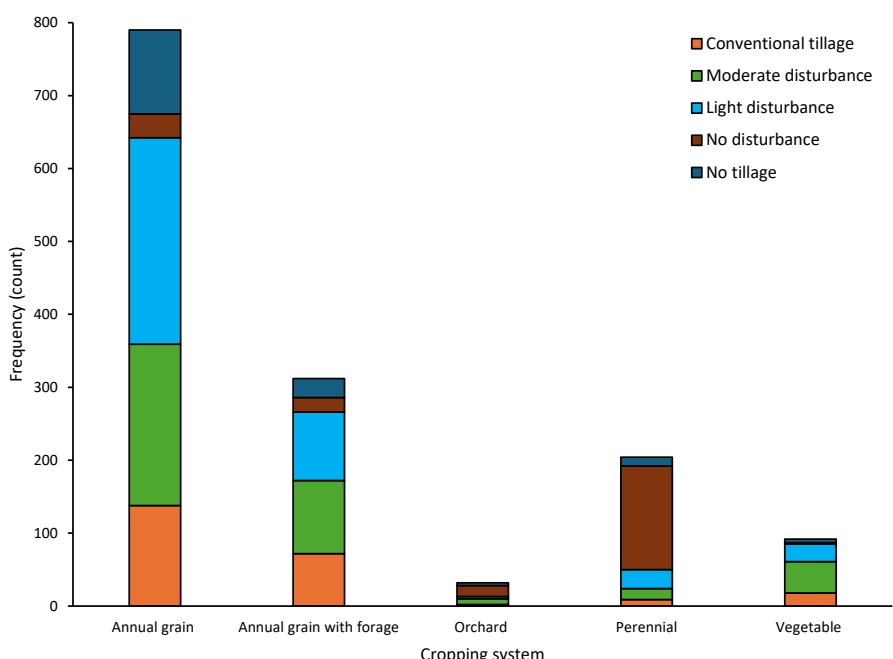

586

**Figure 4** Frequency distribution of soils partitioned by tillage intensity within each cropping system category
sampled in the Ontario Topsoil Sampling Project from 2019 to 2022. Conventional tillage represents the plow tillage
in our study. No disturbance represented little to no soil movement and was associated mainly with pastures and
perennial forages.






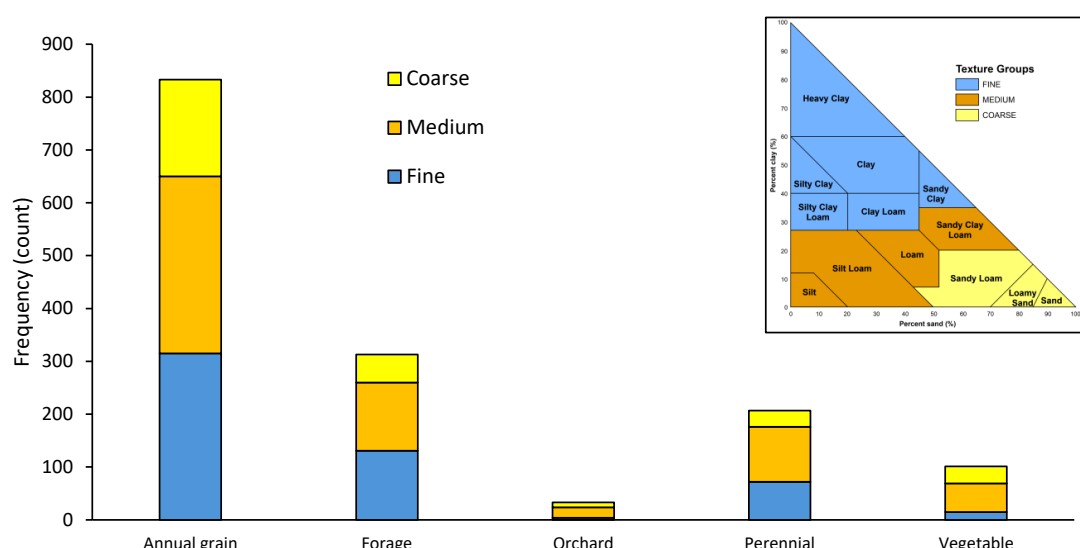



**Figure 5** Frequency distribution of soils partitioned by soil textural classes within each cropping system category
sampled in the Ontario Topsoil Sampling Project from 2019 to 2022.




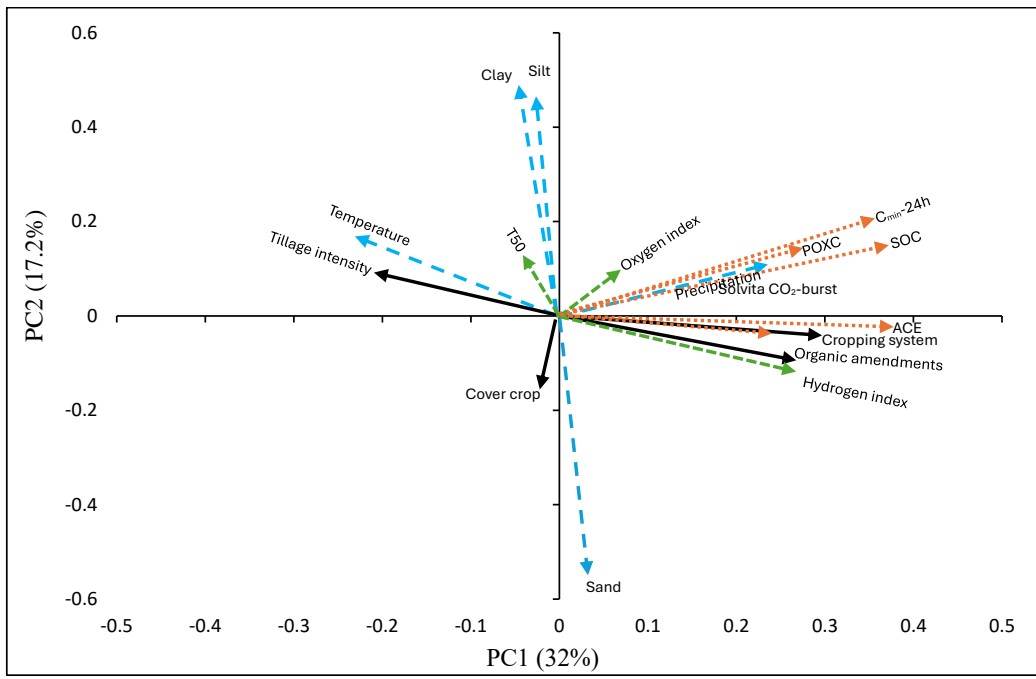

**Figure 6** Principal component analysis (PCA) demonstrating relationships between site characteristics (blue dash-line vectors), management practices (black solid-line vectors), soil C indicators (orange dash-line vectors), and programmed pyrolysis parameters (green dash-line vectors) sampled in the Ontario Topsoil Sampling Project from 2019 (n=151). ACE=autoclaved citrate extractable protein index; $C_{min}$-96h= 96-h carbon mineralization; POXC=permanganate oxidizable carbon; SOC=soil organic carbon; and Solvita $CO_2$-burst= 24 h soil respiration test.