# Peer review of "dataset at landscape scale in Ontario, Canada"

_EGUsphere, 2025_

## Author Comment (AC1)

**Dear Editor**

We thank the reviewer 1 for their feedback on the manuscript. Our response to reviewer comments is in green font. The line numbers correspond to the manuscript file with track changes.

1. **Abstract**

Line 12: It is better to correct sample size to "a dataset of 1490 topsoil samples" or "a dataset of 1490 post-filtered topsoil samples" here.

Authors: Agreed and done.

Line 23: The statement "Remaining soil management variables (cover crop use, tillage intensity, and organic amendments) had less influence" is vague (e.g., "less influence" relative to what?) and conflicts with two key sources. Table 1 shows tillage intensity explains 18.0% of variance in Cmin-96h and 22.3% in ACE; Table 2 further demonstrates tillage significantly affects POXC in fine-textured soils (P=0.0210). The Introduction cites Sun et al. (2023) and Balota et al. (2004), who show reduced tillage and cover crops increase SOC and C mineralization.

I suggest revising vague "less influence" to "weaker influence than texture/cropping systems"

Authors: Thank you. The edit was made at line 24 as per reviewer suggestion.

1. **Introduction**

Line 41: The Introduction cites Weil et al. (2003) for POXC, while Section 2.2 references Moebius-Clune et al. (2016) for POXC methods. The former is the original POXC protocol, the latter is the revised method? Please reconcile the two citations.

Authors: We have deleted the citation "Weil et al. 2003" and have replaced it with "Moebius-Clune et al. (2016)" for clarity at line 43.

1. **Materials and Methods**

Lines 101–102: The original description "Soil samples were collected …… terminated at 30 cm" is ambiguous. Confirm and revise to: "Soil samples were collected from the Ap horizon (agricultural tilled layer). To ensure all samples were restricted to this horizon, sampling was terminated at 30 cm if the Ap horizon exceeded this depth; the median thickness of the sampled Ap horizons across all sites was 25 cm."

Authors: Agreed and done at line 105. "Soil samples were collected from the Ap horizon (agricultural tilled layer). To ensure all samples were restricted to this horizon depth, the sampling depth was terminated at 30 cm if the Ap horizon exceeded this depth. The median thickness of the sampled Ap horizons across all the sites was 25 cm."

Line 112: Overlapping soil texture ranges

The current texture classifications (coarse: 52–94% sand; medium: 2–78% sand; fine: 1–45% sand) have overlapping ranges, leading to ambiguous categorization. Why did the Ref. Moebius-Clune et al., 2016 do like this?

Authors: The categories were designed for soil health scoring functions and not classifying soil textural classes and taxonomy. The reference Moebius-Clune et al. (2016) did not intend these ranges to be exclusive soil textural classes. Having a broad range of soil textural classes allows for smooth transitions of soil health scoring curves.  Therefore, no changes were made to the ranges used to categorize soil textural classes in our study.

Line 130: Explain ACE (n=151) was measured only on the 151 pyrolysis subsamples (representative of full dataset), tell the reader they are the same 151 samples.

Authors: Agreed. This comment was addressed at line 139"As described above, 151 soil samples used to measure ACE protein and pyrolysis parameters were representative of the full dataset and refer to the same set of soil samples."

Line 153-162: Programmed pyrolysis (Section 2.2) is central to the study's novel insights, but key parameters are omitted: helium flow rate (critical for hydrocarbon capture), sample weight precision (±0.1 mg?), and calibration standards used for S1/S2/S3 quantification.

Authors: Thank you for your suggestion. We did not add these parameters as the programmed pyrolysis was conducted as per the standard manufacturer recommended Rock-Eval operating procedures. Since these parameters are well documented in the existing Rock-Eval literature, we believe that including the suggested parameters would not have added any additional information to our study results. To further clarify this comment, we have added that the analyses were performed under standard manufacturer-recommended settings. Line 165 "The standard manufacturer recommended settings (with the helium flow rate at 100 mL/min) were followed to conduct the programmed pyrolysis."

Lines 163–164: the definition of T50 ("temperature at which 50% of the SOC was pyrolyzed") conflicts with the cited reference (Gregorich et al., 2015), which defines T50 for pyrolyzable C (not total SOC).

Authors: We have removed Gregorich et al. (2015) for clarity.

1. **Results and Discussion**

Soil texture explains 60.5% of SOC variance (Table 1)—the largest driver—but the manuscript only attributes this to "inherent soil characteristics" without linking it to specific processes (e.g., clay mineral adsorption, aggregate protection of C). I suggest expanding discussions to: "Soil texture was the strongest driver of SOC (explaining 60.5% of variance; Table 1), primarily due to texture-mediated C stabilization processes: fine-textured soils (high clay content) enhance C retention via clay mineral adsorption (e.g., smectite and illite bind organic molecules) and aggregate protection (microaggregates physically isolate SOC from microbial decomposition). This aligns with Figure 1B, where fine-textured soils had the highest T50 (greater thermal stability), likely due to increased clay-C binding capacity compared to coarse/medium-textured soils."

Authors: Thank you for your comment. We have added the discussion related to soil textural influences on soil C protection in the revised version of the manuscript. Line 204 "Our results of strong dependence of soil texture on SOC confirm the texture mediated soil C stabilization processes. For instance, soil C retention is higher in fine textured than coarse textured soils mainly due to mineral-organic associations and through the formation of microaggregates which physically protect the soil C from microbial decomposition. This finding also aligned with Figure 1 where fine textured soils exhibited greatest T50

values (i.e. indicating greater thermal stability); thus, further highlighting the role of clay induced protection of SOC."

Table 1: "Model $R^2$" values (0.16–0.37) are low, indicating unaccounted variance? Other potential variables (e.g., crop residue input rates, microbial community composition, or historical land use,……)? The manuscript does not address this, leaving readers to question if key drivers were omitted.

Authors: We have added the discussion related to this comment at line 231 in the manuscript. "It is important to note that the model $R^2$ values in our study were relatively low ranging between 0.16 to 0.37 (Table 1). This is not unexpected for agronomic studies on SOC that integrate variable land use management practices and complex biochemical processes. The unexplained variance likely reflects the additional variables which were not captured in our model such as the crop residue inputs, soil microbial community composition, and/or the historical land use intensity. Although these variables are known to influence the SOC dynamics, the data on these factors were not consistently available for all the sites studied. Incorporating these variables in the future research might improve the model $R^2$ values; however, the current results still clearly highlight the importance of soil texture as the major predictor of SOC in our study."

Line 193: "Tillage intensity was not found to be an important predictor" contradicts Table 2, where tillage affects POXC in fine-textured soils (P=0.0210). Qualify this statement.

Authors: We have clarified this comment at line 218 "While tillage intensity was not found to be an important variable impacting soil C indicators when the bulk dataset was used, its effects were detected for some soil C indicators when the data were categorized based on soil textural classes.  In particular, tillage intensity significantly influenced POXC in fine-textured soils (Table 2) suggesting that effects of tillage are more pronounced on labile soil C pools in soils with high clay content than the stable fractions of soil C."

Line 201: Perennials and forages had greatest SOC aligns with Table 3, but Table 3 shows coarse-textured perennial SOC (19.4ab) is not statistically distinct from annual grain (16.9ab). Soften the claim to "greater or comparable concentrations."

Authors: Agreed and done at line 243. "Perennial and forages when grown with annual crops exhibited greater or comparable SOC concentrations relative to the other cropping systems in all soil textural classes (Table 3)."

Line 213: "Diversification of cropping systems... critical for building soil C"—no data on residue inputs to support "quantity/quality of C inputs" as the mechanism. Add a discussion linking cropping systems to measured C inputs (if available) or cite literature on residue differences.

Authors: We have added the discussion related to this comment at line 259 "While our findings suggest that diversification of cropping systems results in greater soil C, we did not measure the quantity and quality of crop residue inputs, which limits our ability to confirm the exact mechanisms of soil C accumulation in our study. Nevertheless, previous studies by King and Blesh (2018), McDaniel and Grandy (2016) have reported that diversifying crop rotations with cover crops, perennials, or forages tend to increase the quantity and biochemical diversity of soil C inputs than the conventional monocultures (e.g. simple annual grain systems)."

Line 253: "In coarse-textured soils, Cmin-96h, POXC... were significantly impacted by tillage"—Figure 2 (boxplots) for these indicators shows overlapping ranges between tillage intensities, weakening the "significant" claim. Discuss effect size alongside statistical significance.

Authors: Thank you for your comment. We would like to clarify that Figure 2 represents the pooled data of soil C indicators across all soil textural classes. The data in Table S1 represents the interaction of cropping system and tillage intensity on soil C indicators within each soil textural class. Because the figure aggregates the data from all the soil textures, the visual overlap in the boxplots does not directly reflect the within texture patterns which were observed in Table S1. Therefore, no edits were made to this sentence to avoid confusion.

Line 260: "Medium and fine textured soils had more interactions than coarse"—no explanation for this pattern (e.g., texture-mediated microbial activity). Add a mechanistic hypothesis.

Authors: Agreed. We have added discussion related to this comment at line 316 "One possible mechanism might be that fine textured soils have high clay content which stabilizes soil C via mineral adsorption and formation of microaggregates. Fine textured soils also promote and support diverse soil microbial communities which play a critical role in supporting the complex microbial mediated soil C transformation processes (Six et al., 2002). In contrast, coarse textured soils have lower surface area, lower water holding capacity, and less soil microbial activity, which perhaps contributed to lesser number of detectable interactions between the management practices on soil C indicators."

Line 262: Inconsistent terminology: "tillage intensity" vs. "tillage treatments" (Line 262). Standardize to "tillage intensity."

Authors: We have revised the sentence at line 324 for clarity.

Line 268: "Vegetable systems... less advanced decomposition" is supported by Figure 3 (high HI, low OI), but Table 3 shows vegetable OI in coarse soils (147) is lower than orchard (203) but not statistically distinct from annual grain (168). Clarify this nuance.

Authors: We have clarified this comment at line 335 "Interestingly, the OI values for vegetable systems were not significantly different than the annual grain systems in coarse textured soils (Table 3), suggesting that the slow decomposition of organic matter in vegetable systems is dependent on the soil texture and agronomic management practices followed."

Figure 6: "Cover crop" is negatively associated with C indicators, but Table 1 shows cover crops explain <5% variance. Discuss why this association is weak (small sample size: n=55?).

Authors: We have addressed this comment at line 404 "Additionally, cover crops were negatively associated with soil C indicators (Figure 6) but explained <5% of the variance in the dataset (Table 1) confirming a very small response of soil C indicators to cover crops (Table S6). It is not entirely clear but might be attributed to a smaller number of observations with cover crops in our study (n=55). The cover crop effects on soil C indicators are largely dependent on the cover crop management factors such as type of cover crop species grown, frequency and duration of cover cropping, planting date, and termination time of cover crops (Blanco-Canqui et al., 2015; Peng et al., 2024). Due to the unavailability of the cover crop management factors in our study, the interpretation of cover crop effect is challenging

but clearly demonstrates that other management factors (cropping system and tillage system) have a greater effect on soil C indicators."

Table 4: "HI vs. ACE (r=0.59***)"—ACE is a labile N/C proxy, but no discussion of why hydrogen-rich labile C correlates with protein. Link this to microbial metabolism of fresh organic matter?

Authors: Agreed. We have added a discussion related to this comment at line 433 "The positive association of ACE with HI confirms that H-rich aliphatic C and protein like N compounds are concomitantly present in fresh organic matter and are co-metabolized by the soil microbes during the early stages of organic matter decomposition."

1. **Conclusion**

It is better to explicitly add methodological limitations, e.g. (1) Tillage intensity categories (e.g., 'light disturbance,' 'no disturbance') relied on producer self-reporting of descriptive terms, with no objective metrics (e.g., plowing depth, number of tillage passes) to standardize classification—introducing bias, as 'light disturbance' may vary by grower interpretation; (2) The small sample size for cover crops (n=55) and orchards (n=33) limits the generalizability of results for these management systems. Future studies should adopt objective tillage measurements and balance sample sizes across management categories to strengthen conclusions.

Authors: Agreed. These limitations were added at line 464 "Additionally, tillage intensity categories in our study were based on subjective producer descriptions and lacked standardized metrics such as tillage depth, number of passes which might have introduced a potential bias in the interpretation of study results. The small sample sizes of cover crops (n=55) and orchards (n=33) limit the generalizability of results for these management practices. Future studies should employ more standardized tillage measurements and ensure a more balanced sample sizes across the agronomic management categories to improve the robustness of study results."

Best regards

Dr. Chao Song

---

## Author Comment (AC2)

**Dear Editor**

We thank the reviewer 2 for their feedback on the manuscript. Our response to reviewer comments is in green font. The line numbers correspond to the manuscript file with track changes.

**General comments**

This study aims to evaluate the impact of agricultural management and environmental variables on soil C indicators, including indicators from thermal analysis. Based on the analysis of 1490 samples, they concluded that soil textured classes and cropping systems had the strongest influence on both quality and stability of soil C indicators, and evidenced interactive effects between cropping systems and tillage intensity. The aim of the subject is very relevant because based on a large dataset at the landscape scale. The experimental approach seems appropriate for answering the scientific questions. I recommend moderate revisions for this paper.

Authors: Thank you.

More information should be given about the delimitation of the geographical area. In general, several results warrant further explanation and discussion. The authors should further elaborate on the limitations of the approaches and adopt a more critical perspective towards them. The part concerning the PCA interpretation should be improved.

Authors: Thank you. We have addressed all the reviewer comments and believe that these revisions have significantly improved the manuscript.

**Detailed comments**

*Introduction*

Line 32-36: "Soil C" à do the authors mean soil C dynamic or turnover or stability or …? Soil C alone is not a parameter. The parameter or process meant in this paragraph should be specified.

Authors: Agreed and clarified at line 33 by adding soil C dynamics.

Line 34: "Soil organic C (SOC)" content

Authors: We have added "content or concentration" at line 36. Both the terms SOC content or SOC concentration are accepted in the literature, and we used both the terms here to be consistent with published research and to avoid confusion.

Line 81-82: I am not sure to understand if the research gap is due to the difficulty to observe any effect of the previously cited factors on C stability, or if it is due to the difficulty to identify the main factors affecting C stability. I suggest reformulating this sentence.

Authors: Agreed. Sentence revised at line 83 "Yet, it remains uncertain which management or environmental variables exert the strongest influence on soil C stability under the complexity of agricultural fields."

Line 91: again: "soil C " storage and stability?

Authors: Agreed and edit made at line 94.

*Material and method*

Line 97: Since the study highlights that the experiment was done at a landscape scale, the authors should specify the geographical area of the project. It may be important to have an idea if the scale of the area (regional, national or multi-national scale?) and if the studied parameters (e.g. cropping systems, tillage intensity, use of organic amendment) are representative of this geographical area.

Authors: We have clarified this comment at line 100 "The soil samples for this project were collected from multiple locations throughout southern Ontario."
Line 112 "The agricultural management factors identified in our study were consistent with the commonly adopted practices by the growers and were representative of the geographical area."

Line 129: the authors should precise what "QA/QC analysis" refers to.

Authors: We have clarified this at line 137 "After quality assurance and quality control (QA/QC) analysis"

Line 159: Were the CO emissions also included in the S3?

Authors: Yes. We did not add this edit to the manuscript to avoid potential confusion.

Line 163: I understand that T50 corresponds to 50% of C emissions under all S1, S2 and S3 signals. If not, this should be specified.

Authors: We have clarified this comment at line 176 " It is important to note that T50 is measured under S1 and S2 only."

Line 167: if "variance component analysis" are made, the data should follow normal distributions. It would be relevant to precise here if the normality and homoskedasticity were tested for the data. Unless the authors consider that the sample sizes are large enough to override the normality of the data...

Authors: We have clarified this at line 180 "Prior to conducting the variance component analysis, the assumptions of normality were assessed. Given the relatively large sample sizes for most of the variables studied, variance component estimates were considered robust to minor deviations from normality."

*Results and discussion*

Line 190: it is not clear what "75.2%" refers to.

Authors: We have reorganized this sentence for clarity. Line 214" Soil texture also explained a large amount of variance (75.2%) in the thermal-based parameters of soil C (Table 1)."

Table 1: it is surprising that the carbon inputs, such as cover crops (if plant residues remain on the field) and organic amendments, have so low effects on SOC and on the indicators of C stability, when compared to cropping system and soil texture. Indeed, organic matters amendment is expected to increase labile C content and hence, decrease the overall stability of C. These results are, at least to me, very surprising and therefore interesting. I suggest developing a bit more a critical discussion about these results and the approach, and eventually present hypothesis of explanation. An idea to complete this approach, could be to make a partitioning of the variances within each textural class.

Authors: We have added some discussion based on this and reviewer 1 comments. The results and discussion section was thoroughly revised to address both reviewer comments. This reviewer comment was addressed at line 223 "The least amount of variance in soil C indicators was explained by MAT, use of cover crops, and organic amendments (Table 1), suggesting a minor influence of these factors on soil C variability in our study. One possible explanation for this result could be that cover crop and organic amendment effects varies with soil type, climatic conditions, and baseline fertility which might have potentially masked their overall impact on soil C indicators in our multi-site study."

We, however, do not agree with the reviewer regarding partitioning of variances within each textural class. We believe that the current analyses adequately address the study results. Given the structure of our data, further partitioning of variances by soil textural class might result in misleading interpretations of our dataset and would not provide any additional information. We have, however, focused on providing the potential mechanisms based on the study results and have revised the results and discussion throughout the manuscript.

Table 2: is there any reason why annual temperature and mean precipitation are not tested in this table? Furthermore, the results evidence statistical differences but ignore tendencies that may also be interesting to investigate.

Authors: Thank you for your comment. Annual temperature and mean precipitation were not tested in Table 2 because the focus was on the agronomic management factors and their interactions. Nevertheless, we have acknowledged that environmental variables like annual precipitation and temperature are important predictors of soil C dynamics and have thoroughly discussed this in the manuscript. While tendencies might be interesting to investigate, our analysis focused on statistically supported differences to ensure the reliability of conclusions. We did not add discussion on trends and tendencies to avoid overinterpretation of study results.

Table 3: in the legend, the authors should remind which statistical differences are represented by the letters. As for tables 1 & 2, the authors should add what T50 refers to.

Authors: Agreed and done.

Line 210: "the lowest soil C" content? Stability? Please precise which indicator are referred to.

Authors: Sentence revised for clarity at line 255.

Lines 210-211: I guess that many types of management exist to produce vegetable and annual grain systems. They are maybe not always intensively managed. I guess that this could be a critical point you

could add to the discussion. It would be interesting to have a bit more information about how the fields of the project were managed and if they were all managed the same way within each cropping system.

Authors: We agree with the reviewer's comment that there are nuances within each of the cropping systems used in the analysis of the data, and that management is not consistent across all farms within each cropping system. In fact, this is the reason additional parameters were included in the analysis (i.e., tillage, cover crops, organic amendments). The statement in question is meant as a general statement to reflect the fact that annual cropping systems and vegetable production systems, when compared to perennial cropping systems, are in general more intensive, as supported by the cited literature. Therefore, we did not add any more details about the management within each cropping system.

Line 219: Forage doesn't have the lowest T50 across all soil textures in table 3; vegetable and perennial have lower values than forage for medium-textured soils, and orchard has lower values for fine-textured soils.

Authors: We have revised this sentence. We have removed Table 3 from this sentence for clarity. This result was based on average values across all soil textures. Line 269 "While T50 was not statistically different due to cropping system and tillage practices in all soil textures (Table 2), annual grain had the highest whereas forage had the lowest T50 when averaged across all soil textures (Figure 1a)."

Lines 226-228: This result is interesting, and therefore it would be relevant to develop the discussion about the influence of the texture on thermal stability. There may be a link to do with the higher organo-mineral associations in clay than in sandy soils. It would also be relevant to precise if other studies also found this same result.

Authors: Thank you. We have added a discussion to address the reviewer comment at line 279 "It is likely related to the greater organo-mineral associations in clay rich fine textured soils than the coarse textured soils, which contributed to the protection of soil organic matter from microbial decomposition and increase its thermal stability.  Previous studies by Simkovic et al. (2025) and Stoner et al. (2023) have also confirmed a positive relationship between clay content and stabilization of soil organic matter."

Figure 2: In the legend, the authors wrote that the data from the other cropping systems are not shown because of the insufficient number observations. If these number of observations were not a problem for the other tables and figures, why should they be a problem for this figure? It should especially not be a problem when presented as boxplots.

Authors: The reason for not including the data from other cropping systems in Figure 2 is that the number of observations for the other cropping systems was low and unevenly distributed, which could make the boxplots difficult to interpret and misleading.  In the other tables and figures, the data were presented as summary statistics or treatment means which are less sensitive to uneven sample size. Boxplots visually help to understand the distribution of observations and including cropping systems with very few observations would have given a false impression of variability. Therefore, we only used cropping systems with sufficient number of observations for meaningful comparisons.

Line 251: I guess you mean "Table 2" instead of "Figure 2".

Authors: Yes. Thank you. It was a typo and fixed in the manuscript text.

Lines 247-264: in this paragraph, there are several mentions of the Table S1. It could be relevant to add this table in the main tables.

Authors: Thanks for your suggestion. While we understand that information in Table S1 supports the main text, we did not include it in the main manuscript tables. We believe that including Table S1 as main table would disrupt the flow and clarity of the manuscript. We have therefore kept it as a supplementary table while ensuring that all the relevant information is appropriately referenced and discussed in the main manuscript text.

Line 269: The HI of vegetable systems doesn't seem different than those of orchard systems and thus, this figure doesn't show that the organic matter of vegetable systems mainly consists

Authors: To address this comment and for clarity, we have inserted a sentence at line 332 "While the visual representation of HI vs OI between both systems (i.e. vegetable and orchards) may appear similar due to variability and sample size, the underlying data distribution supports our interpretation (Figure 3)."

Lines 274-287: This paragraph, as well as figures 4 and 5 should be transferred in material and method. In the discussion should only appear how this distribution may affect the results. Figures 4 and 5 could as well be transferred as supplementary materials.

Authors: We thank the reviewer for their suggestion. However, we believe that this paragraph and Figures 4 and 5 are essential components of the results and discussion section. This paragraph and Figures provide core results and key contextual information that directly supports the interpretation of our study findings and explains the variability observed. One possible explanation of the observed results is related to the frequency distribution of the data, which is presented in Figures 4 and 5. Moving this paragraph and figures to methods or supplementary section would limit the understanding of the study results. Therefore, we prefer not to move them to methods or supplementary section of the manuscript.

Line 289: The authors should introduce in the main text how the PCA was done and with which indicators and parameters.

Authors: Agreed. We have provided details about this comment in the main text at line 192 ". In addition to the site characteristics and pedoclimatic conditions, the variables included in the PCA were SOC concentration, POXC, $C_{min}$-24h, Solvita $CO_2$-burst, ACE, HI, OI, and T50. The first two PCs were selected based on scree plots and the eigenvalues of the soil C indicators were used to create the biplots to better understand the interdependence among the soil C indicators and how they interacted with the site characteristics and pedoclimatic conditions."

Line 290: did the other studies used also the same parameters and indicators? If yes, on which type of data did they use it? It is a bit tricky to compare variances from different datasets…

Authors: We have deleted the "which is consistent with variance explained in other studies (Liptzin et al., 2022)." from the sentence for clarity.

Lines 294-296: It would be interesting to have an explanation of why the different approaches of variance analysis and PCA do not show the same results concerning the influence of weather (temperature and humidity) on C indicators.

Authors: Thank you for this comment. We have clarified this comment at line 363 "It is important to note that PCA and variance component analysis differ in both the statistical structure and objectives, which perhaps led to differences in the results between both approaches. For instance, variance analysis evaluates the independent effect of each predictor variable on soil C indicators, whereas PCA simultaneously assesses the covariance among the multiple soil C indicators."

Lines 307-314: In the PCA, the arrows from the soil texture are orthogonal to the arrows from the soil C indicators, thereby showing no correlations. Therefore, I disagree with the results presented in this paragraph.

Authors: We have significantly revised this paragraph to address reviewer comments and for clarity. Lines 378 "Furthermore, silt and clay content were clustered together in PCA on one side of the second axis whereas sand content was positioned on the opposite side of the second axis (Figure 6). This result was consistent with the well-established associations between soil C dynamics and soil texture where soils rich in clay content have higher C retention capacity than coarse textured soil (von Lutzow et al., 2006). Accordingly, the positive loading displayed by the clay and silt rich soils on the second axis corresponds to greater values of soil C indicators observed in our study. Although important, the relationship between soil texture and soil C indicators (particularly POXC and respiration) has not been explored enough in the literature (Nunes et al., 2020; Sinsabaugh et al., 2008)."

Line 315: idem, HI and the C indicators are orthogonal to the cover crops and thus, the PCA does not show any correlation between them. In addition, the cover crop reflects a very low contribution in the PCA analysis (small arrow size).

Authors: To avoid confusion and for clarity, we have removed "cover crops" from this sentence. We have only focused and discussed the results of tillage intensity and how it impacts soil C dynamics in this paragraph.